# Spatiotemporal Variation and Predictors of Unsuppressed Viral Load among HIV-Positive Men and Women in Rural and Peri-Urban KwaZulu-Natal, South Africa

**DOI:** 10.3390/tropicalmed7090232

**Published:** 2022-09-06

**Authors:** Adenike O. Soogun, Ayesha B. M. Kharsany, Temesgen Zewotir, Delia North, Ebenezer Ogunsakin, Perry Rakgoale

**Affiliations:** 1Department of Statistics, School of Mathematics, Statistics and Computer Science, College of Agriculture Engineering and Science, University of KwaZulu-Natal, Durban 4001, South Africa; 2Centre for the AIDS Programme of Research in South Africa (CAPRISA), Doris Duke Medical Research Institute, Nelson R. Mandela School of Medicine, University of KwaZulu-Natal, Durban 4001, South Africa; 3School of Laboratory Medicine & Medical Science, Nelson R Mandela School of Medicine, University of KwaZulu-Natal, Durban 4001, South Africa; 4Discipline of Public Health Medicine, University of KwaZulu-Natal, Durban 4001, South Africa; 5Department of Geography, School of Agriculture, Earth, and Environmental Science, University of KwaZulu-Natal, Durban 4001, South Africa

**Keywords:** unsuppressed HIV viral load, Bayesian, spatial effect, geoadditive model, integrated nested Laplace approximation (INLA), non-linear effect, small enumeration area, South Africa

## Abstract

Unsuppressed HIV viral load is an important marker of sustained HIV transmission. We investigated the prevalence, predictors, and high-risk areas of unsuppressed HIV viral load among HIV-positive men and women. Unsuppressed HIV viral load was defined as viral load of ≥400 copies/mL. Data from the HIV Incidence District Surveillance System (HIPSS), a longitudinal study undertaken between June 2014 to June 2016 among men and women aged 15–49 years in rural and peri-urban KwaZulu-Natal, South Africa, were analysed. A Bayesian geoadditive regression model which includes a spatial effect for a small enumeration area was applied using an integrated nested Laplace approximation (INLA) function while accounting for unobserved factors, non-linear effects of selected continuous variables, and spatial autocorrelation. The prevalence of unsuppressed HIV viral load was 46.1% [95% CI: 44.3–47.8]. Predictors of unsuppressed HIV viral load were incomplete high school education, being away from home for more than a month, alcohol consumption, no prior knowledge of HIV status, not ever tested for HIV, not on antiretroviral therapy (ART), on tuberculosis (TB) medication, having two or more sexual partners in the last 12 months, and having a CD4 cell count of <350 cells/μL. A positive non-linear effect of age, household size, and the number of lifetime HIV tests was identified. The higher-risk pattern of unsuppressed HIV viral load occurred in the northwest and northeast of the study area. Identifying predictors of unsuppressed viral load in a localized geographic area and information from spatial risk maps are important for targeted prevention and treatment programs to reduce the transmission of HIV.

## 1. Introduction

Globally in 2020, 36 million adults 15 years and older were living with human immunodeficiency virus (HIV) and 1.5 million became newly infected, whilst 680,000 deaths from acquired immunodeficiency syndrome (AIDS)-related illness were reported [1,2]. South Africa contributes to about 22% of the global HIV burden [3], with the province of KwaZulu-Natal (KZN) having the highest prevalence and incidence [4,5,6]. Therefore, surveillance for HIV infection is important in this region where prevalence and incidence remain persistently high [5,6,7].

Since 2010, South Africa as a country has made substantial progress to improve HIV testing services to enhance knowledge of HIV status [8], improve uptake of antiretroviral treatment (ART) [9,10], prevent mother-to-child transmission (PMTCT) of HIV [11,12], improve uptake of voluntary medical male circumcision (VMMC) [13,14], and provision of HIV pre- and post-exposure prophylaxis (PrEP and PEP respectively) to reduce HIV transmission potential and improve HIV-related morbidity and mortality and increase life expectancy [15]. The rapid scale-up of ART has an estimated 5 of the 7.5 million adults living with HIV receiving ART [3,4]. South Africa has also been signatory to the Joint United Nations Programme on HIV/AIDS (UNAIDS) 90-90-90 HIV treatment targets [16], including the revised targets of 95-95-95, with the goal of achieving a composite measure of 86% viral suppression at a community level [1]. However, these targets have not been met [17,18,19] and community viral load remains an important key driver of the epidemic [20,21,22], especially in regions with suboptimal coverage of ART [4].

HIV viral load <400 copies/mL among people living with HIV (PLHIV) has been recommended as a threshold measure for evaluating the effectiveness of ART at the individual and community levels [23,24,25,26]. ART crucially prevents viral replication and preserves and boosts immunological well-being [15,27]. Viral suppression is a marker of HIV treatment success [21], thus reducing onward transmission of HIV [6,20,21]. For public benefit, South Africa has also adopted the Universal Test and Treat (UTT) strategy to fast-track the goal to achieve viral suppression [28]. However, several factors are associated with a lack of achieving HIV viral suppression [29,30,31,32] and individuals with unsuppressed HIV viral load (that is, those with HIV viral load ≥400 copies per mL), continue to sustain the epidemic.

Extensive geographic variation exists in the spatial distribution of HIV epidemiological measures such as prevalence, incidence, and viral load levels across sub-Saharan Africa [33,34,35,36,37,38,39]. Therefore, small-area localized-based modelling approaches are important to understand these measures and identify key populations, risk areas, risk factors, and targeted interventions to impact and reduce the HIV burden [7,33]. Likewise, in modelling spatial data, the presence of spatial autocorrelation between observations and residuals must be considered to avoid misleading results. Thus, over time, spatial analyses aided by regression models have contributed to advancing our understanding in the distribution of HIV for targeted interventions [35,37,39].

Using conventional statistical approaches, with a focus on individuals attending health services for ART, several studies have highlighted socio-demographic, behavioural, clinical, epidemiological, and geographical factors as predictors of unsuppressed HIV viral load [29,30,31,32,40,41,42]. A study from Kenya, at a country level, used the Bayesian framework in modelling the spatial trends and factors associated with viral suppression [43], whilst a study on women from KZN, South Africa, assessed unsuppressed HIV viral load using population-based data and multilevel modelling [29]. Several studies have recommended the utility of spatial analyses to accelerate targeted responses towards achieving the UNAIDS targets [29,44]. Spatiotemporal analysis can be used for disease surveillance over time and space to identify predictors of unsuppressed viral load among men and women in this HIV-hyperendemic area. There exists a substantial gap in the analysis of predictors of unsuppressed viral load from previous studies, where space and time were not accounted for [29,31,40,41]. Building on recent developments in computation and analyses, approaches have necessitated the use of advanced statistical techniques, where the Bayesian method has become the mainstream method in spatial statistics modelling and epidemiological inferences to accurately inform public health practitioners [45,46,47,48]. Importantly, focusing on hotspots and disease mapping helps prioritize the allocation of limited resources [7,33].

We investigated the prevalence, predictors, and high-risk areas of unsuppressed HIV viral load among HIV-positive men and women using a Bayesian geoadditive regression model while accounting for unobserved factors, time, non-linear effects of selected continuous variables, and spatial autocorrelation using data from a population-based study.

## 2. Materials and Methods

### 2.1. Study Setting and Population

This analysis was based on data obtained from men and women aged 15 to 49 years enrolled in the HIV Incidence Provincial Surveillance System (HIPSS), a population-based study in rural Vulindlela and the adjacent peri-urban Greater Edendale areas in KwaZulu-Natal, South Africa (Figure 1). Briefly, the study was designed to measure HIV prevalence and incidence following the programmatic scale-up of HIV prevention and treatment efforts in a “real world” non-trial setting [49]. From a total of 600 enumeration areas (EAs), all 591 EAs with more than 50 households were systematically selected at random, of which 221 and 203 EAs were randomly drawn for the 2014 and 2015 surveys, respectively. The EAs are located between 29°39′ south and 30°17 east of KwaZulu-Natal, which covers a total of 33 wards in the Msunduzi and a part of the uMngeni municipalities area of the uMgungundlovu District.

### 2.2. Study Design and Data

Two sequential cross-sectional surveys were undertaken from 11 June 2014 to 18 June 2015 (2014 Survey) and 8 July 2015 to 7 June 2016 (2015 Survey). Households were geo-referenced and randomly selected using two-stage random sampling methods and one individual per household within the age range of 15–49 years was randomly selected and enrolled following written informed consent. Questionnaires were administered to obtain household- and individual-level data on demographics, socio-economic status, and health-related information. All enrolled participants provided peripheral blood samples for laboratory measurements for HIV. Across the two surveys, a total of 20,048 individuals were enrolled and 7839 were diagnosed HIV-positive and 7824 had HIV viral load measurements. Fifteen (15) participants with missing viral load data were excluded from the analysis, resulting in a final sample of n = 7824. The primary outcome was unsuppressed HIV viral load defined as HIV viral load of ≥400 copies/mL. The study protocol and procedures have been described in detail elsewhere [49].

### 2.3. Laboratory Measurements

HIV antibodies were measured using the 4th-generation HIV enzyme immunoassay Vironostika HIV Uniform II Antigen/Antibody micro-enzyme-linked immunosorbent assay (ELISA) system (BioMérieux, Marcy l’Étoile, France) and Elecsys HIV 1/2 combi PT assay (Roche Diagnostics, Penzberg, Germany) [50]. All HIV antibody-positive samples were confirmed by Western blot (Western Blot–HIV-1 kit. Bio-Rad Laboratories, Redmond, WA, USA) [50,51]. HIV viral load testing was done using assays with high levels of sensitivity and specificity to minimize misclassification (COBAS AmpliPrep/COBAS TaqMan HIV-1 version 2.0 assay (Roche Diagnostics, Penzberg, Germany) as per manufacturers’ instructions [51,52].

### 2.4. Study Variables

The dependent variable was HIV viral load status which was dichotomized as a binary outcome defined as:(1)$$\Phi_{ij}=\{\begin{array}{c} 1\;\;\;\;\;\;viral\;load\;\;\;\geq 400\;copies/ml\;\;\left(unsuppressed\right)\\ 0\;\;\;\;\;viral\;load<400\;copies/ml\;\;\;\;\;\;\;\;\;\left(suppressed\right) \end{array}$$

This threshold was used because the potential risk for HIV transmission had been shown to be absent at HIV viral load of ≤400 copies/mL and is in accordance with the South Africa National Department of Health ART treatment guidelines [23,24,25,26]. Viral suppression calculation and definitions were based on the composite viral suppression of all PLHIV, irrespective of being on ART or not [1,2,16]. To avoid covariate selection bias, the potential covariates used were socio-demographic, behavioural factors, perception, and knowledge of HIV testing and status, exposure to and treatment for tuberculosis (TB) and sexually transmitted infection (STI), and geographic and clinical characteristics, based on initial exploratory data from previous study [53] and findings from the literature (Appendix A).

### 2.5. Statistical Analysis

A unique identification number was allocated to link the household, participant’s questionnaire, and laboratory data [5,6,17]. A spatial autocorrelation test was used to assess spatial variation and quantify the correlation of the observed and residual prevalence. To assess this spatial variation, both Geary’s C-test and convoluted spatial structure was utilized. Firstly, Geary’s C-test was used due to its consistent power, the localized nature of this study area, and the test’s sensitivity power to differentiate smaller spatial areas as compared to Global Moran’s I [54,55]. Geary’s C-test was applied on the dependent variable, spatially aggregated over the EAs. Secondly, from Bayesian inference, convoluted spatial structure was used, which assumes the decomposition of spatial effect into two components: structured and unstructured spatial effects due to dependency and similarities in geographic proximity [56]. The unstructured spatial effect accounts for the unobserved covariates that are intrinsic within the enumeration area, for example, structural and cultural practices, climate conditions, etc., while the structured spatial effect accounts for any spatially varying unobserved covariates among the enumeration areas.

To account for the complex multilevel sampling design, weighted percentage and frequency were used to summarize and describe the study characteristics and prevalence of unsuppressed HIV viral load. Weighted proportion of unsuppressed HIV viral load was estimated using the Taylor series method; percentages, 95% confidence intervals (CIs), and *p*-values results are shown. The variance inflation factor (VIF) was used to check for collinearity among the continuous independent variables; in all variables with VIF < 4, multicollinearity was assumed to be significantly present and these variables were excluded [57,58]. To test the pairwise strength of association between individual covariates and the dependent variable, we conducted a weighted bivariate analysis using the Rao–Scott chi-square test. Covariate variables that were independently associated with the dependent variable at a 5% level of significance were included in the multivariate model. For the classical bivariate analysis, odds ratios (OR) were reported as a point estimate with their corresponding 95% CIs and *p*-value. Three multivariate models were compared to understand the sensitivity of the prior choice and accuracy of the model, namely the generalized additive model (GAM), structured additive model (SAM), and unstructured additive model (UAM). Model comparison was measured using the deviance inclusion criterion (DIC), and the smallest value of the DIC was considered best-fit and was further used in the Bayesian geoadditive model [59,60]. Our study employed the Bayesian approach; thus, the DIC deployed for model selection is the Bayesian information criterion (BIC) [59]. The Bayesian approach is superior when models contain residual spatial dependence because this approach offers robust and more consistent estimates via capturing the uncertainty that is intricate in parameter estimation [59,60]. The posterior-adjusted odds ratios (AOR) with their corresponding 95% credible intervals (CrI) and fixed effect were reported to identify predictors and high-risk areas of unsuppressed viral load. To test the significance of individual predictors, we used the 95% CrI; if these intervals contained the number zero (0), this means the parameter is not significant; otherwise, it is significant.

The non-linear effect of all continuous variables was also examined. The structured additive regression model offered a flexible extension to standard regression models by enabling simultaneous modelling of possible non-linear effect of continuous covariates, spatial correlation, and heterogeneity while estimating fixed effects of categorical and continuous observed variables [61,62]. Current age, household size, and number of lifetime HIV tests were the only non-linear continuous covariates considered in the fitted model, while other continuous covariates were included as linear fixed effects. To examine the accuracy of the non-linear effect and the spatial effects on the models, the smooth-term variance component (SVC) with the smallest value was considered accurate [63]. The observed prevalence mapping and development of the study area shapefiles were carried out in ArcGIS version 10.6.1 (Environmental System Research Institute (ESRI), Redlands, CA, USA). Descriptive and bivariate statistical analysis was carried out in SAS (SAS Institute, Cary, North Carolina) version 9.4. Bayesian geoadditive modelling, spatial effect mapping, and non-linear plots were carried out in R software (version 4.0).

### 2.6. Statistical Model Formulation

To account for the spatial variation and connotation of the individual-level predictors with the outcome of interest, three models were considered for comparison. The spatial analysis deployed in the current study is built on the notion of the structured additive model (SAM) considering a binary response. Suppose the outcome of interest (HIV viral load status) of the $j^{th}$ individual residing in the $i^{th}$ enumeration area is represented as $\gamma_{ij.}$ Meanwhile, it is assumed that the distribution of this outcome is linked to the exponential family, which can also be linked to the linear predictor ϕ, given as: ψ=g(ϕ) such that $\phi =\varsigma^{T}\lambda$. In this case, g denotes a response function and λ implies a regression parameter. In the case of our study, existing covariates comprise spatial, continuous, and categorical, which can then be painstaking within the context of the generalized additive model (GAM) [64,65,66]. After the SAM, the predictor $\phi_{ij}$ can be described as follows:(2)$$\phi_{ij}=\Phi_{2}+\sum_{j=1}^{m}f_{ij}\left(\varsigma_{ij}\right)+f_{spatial}\left(S_{i}\right)+\tau_{ij}\lambda$$
where $\Phi_{2}$ denotes the year of study, *f* represents the non-linear unidentified smooth function of the covariate age (ς), λ is the regression coefficient for the categorical covariates (τ), and $f_{spatial}$ denotes spatial effect (s). To consider spatial heterogeneity, from Equation (2), the spatial effect was introduced and was further split into a spatially correlated (structured) and an unstructured (uncorrelated) effect. Thus:(3)$$f_{spatial}\left(S_{i}\right)=f_{struc}\left(S_{i}\right)+f_{unstruc}\left(S_{i}\right)$$

Based on the above two specifications, the current study fitted three different models to understand the impact of the various explanatory variables. The spatial modelling was performed in the R statistical software, version 4.1.1, using the ‘INLA’ package [67,68,69]. The analysis was carried out in three stages. The first approach modelled the space covariate effects and the non-linear effects of the covariate age, number of lifetime HIV tests, (NLHIVtest), and household size (hhs). This model does not consider the spatially structured and spatially unstructured random effects.

The generalized linear modelling approach associated the mean response and the potential k predictors (X1,……….., X_ijm_)^T^ in such a way so that a function of $\eta_{ij}$ is equal to the linear combination of predictors Equation (4). In this case, $\eta_{ij}$ is the probability of being virally unsuppressed for the *j*th individual in the *i*th enumeration area. For the Bernoulli outcome, the logit is a commonly used link function defined as specified in Equation (4) below.
(4)$$\log\left(\frac{\eta_{ij}}{1-\eta_{ij}}\right)=\Phi_{2}+\sum_{m=1}^{k}\beta_{m}x_{ijm}$$
where ${\left\{\beta_{m}\right\}}_{m=1}^{k}$ denotes the regression coefficients for the observed predictors. However, often there exist numerous unobserved predictors that cannot be considered in the model specified above. Some of them are inherent within the district and cause correlation among individuals from the same province due to standard cultural practices, attitudes, climate, and health interventions. Additionally, the other unobserved predictors vary spatially across the district and cause spatial cross-correlation, which is strictly described as dependence due to geographical closeness. Therefore, by introducing a spatially unstructured random effect ($\nu_{i}$) for the first part (that is, within province correlation) and a spatially structured random effect ($s_{i}$) for the later part (that is, spatial autocorrelation) in Model 1, one can capture the effects of these unobserved predictors. First, with unstructured random effects $\nu_{i}$ at a district level, Model 1, represented by Equation (4), can be expanded to Model 2, denoted by Equation (5), with the logit link function as follows:(5)$$\log\left(\frac{\eta_{ij}}{1-\eta_{ij}}\right)=\Phi_{2}+\sum_{m=1}^{k}\beta_{m}x_{ijm}+f\left(age\right)+f\left(hhs\right)+f\left(NLHIVtest\right)+\nu_{i}$$

Additionally, the probabilistic nature of the structured spatial random effect is usually described by the independent average density with mean zero and variance $\sigma_{v}^{2}$. Therefore, with the structured spatial random effect denoted by $s_{i}$, we thus expanded Equation (4) to Equation (6), denoted Model 3 with logit link function as follows:(6)$$\log\left(\frac{\eta_{ij}}{1-\eta_{ij}}\right)=\Phi_{2}+\sum_{m=1}^{k}\beta_{m}x_{ijm}+\xi_{i}+f\left(age\right)+f\left(hhs\right)+f\left(NLHIVtest\right)+s_{i}$$

However, Model 3 assumes only a spatially structured component. The unstructured random effect within area *i* is modelled as spatial correlation, leading to a biased estimate of the model parameters. To overwhelm this limitation, the Besag–York–Mollie (BYM) model provided an alternative measure by allowing a convolution of an unstructured and a structured spatial component as a random effect, which resulted in the formulation of Equation (7), denoted final Model, as follows:(7)$$\log\left(\frac{\eta_{ij}}{1-\eta_{ij}}\right)=\Phi_{2}+\sum_{m=1}^{k}\beta_{m}x_{ijm}+\xi_{i}+f\left(age\right)+f\left(hhs\right)+f\left(NLHIVtest\right)+\nu_{i}+s_{i}$$

For the full Bayesian inference, spatial smoothness priors for the spatially structured component are employed in modelling, namely a Gaussian Markov random field (GMRF), and a conditional autoregressive (CAR) model [67,70,71,72,73]. Suppose that the index s ∈(1, 2..., S) represents the geographically connected regions. Two closely neighbouring locations are unlikely to be independent, which breaks the assumptions of most conventional statistical analyses. To relax these assumptions of spatial correlation, within a small enumeration area, a suitable approach must be used to circumvent these shortfalls [74]. In our case, two EAs *s* and *s^’^* are neighbours, as they share a common boundary. The spatial smoothness prior for the function evaluation f(s) is given by:(8)$$f\left(s\right)/f\left(s\prime \right),s\neq s\prime ,\tau^{2}∼N\left(\frac{1}{N_{s}}\sum f_{str}\left(s\prime \right)\frac{\tau^{2}}{N_{s}}\right)$$
where $N_{s}$ is the number of neighbors’ EAs and the (conditional) mean of $f_{str}\left(s\right)$ denoting an average of function assessments $f_{str}\left(s\right)$ of neighboring EAs, while $\tau^{2}$ is the variance parameter. For a spatially unstructured effect, $f_{unstr}$ an i.i.d Gaussian prior $f_{unstr}\left(s\right)$ are i.i.d was assigned and denoted as:(9)$$f\left(s\right)|\tau_{unstr}^{2}∼N\left(0,\tau_{unstr}^{2}\right)$$

Thus, we applied the integrated nested Laplace approximation (INLA), which imitates a Bayesian Monte Carlo Markov Chain (MCMC) [48] via R package R-INLA [47,48,67,68,69], a computationally intensive and faster Bayesian approximation. INLA is an alternative Bayesian estimation method that computes approximations of posterior marginal distributions for latent Gaussian models, and it provides accurate estimates of the integrals through a Laplace approximation.

To establish the statistical meaning of the fitted models, it is essential to evaluate the parameter values. For the present study, we deployed a Bayesian method for parameter estimation. There are two important reasons for this choice. First, the Bayesian method exemplifies uncertainty at all phases in the modelling development. Furthermore, the Bayesian approach is flexible because it permits the development of complex models. Hence, under Bayes’ theorem, the parameters’ posterior distribution is proportional to the product of the likelihood and parameters’ prior distribution: Posterior ∝ Prior × Likelihood.

Hence, this posterior distribution is then deployed for inference. In order to use prior evidence to identify these parameters, the Bayesian method, which can maximize statistical power by using all the information available while considering uncertainty in both the data and parameter estimates, ensures that the prior evidence is used to avoid the identifiability problem. Therefore, a Bayesian approach can estimate all unknown parameters in the model. Thus, in a Bayesian technique, unknown functions $f_{j},j=1,....k$, $f_{str}$, $f_{unstr}$ and parameters λ, and the variance parameter $\sigma^{2}$ are considered random variables and have to be accompanied by proper prior assumptions. Bayesian inference was based on the posterior and was carried out using recent INLA methods. The main principle behind INLA lies in approximating the posterior marginals of the wide range of Bayesian hierarchical models. Let *y* denote the observed data points, ϕ be the vector of all the latent Gaussian variables, and θ denote the vector of hyperparameters. Assuming conditional independence, the likelihood of the *n* observations *y* is given by:(10)$$\pi \left(y_{i}|\phi ,\theta \right)=\prod_{i=1}^{n}\pi \left(y_{i}|\phi_{i},\theta \right)$$
and assuming a multivariate Gaussian prior on ϕ with zero mean and precision matrix B(θ), the density function of the latent effects is given by
(11)$$\pi \left(\phi |\theta \right)=\frac{1}{\sqrt{2\pi }}{\left|B\left(\theta \right)\right|}^{1/2}\exp\left(-\frac{1}{2}\phi^{T}B(\theta )\phi \right)$$
where |·| denotes the matrix determinant. The property of ϕ are that they are conditionally independent such that B(θ) is a sparse matrix which allows inference with Gaussian Markov random fields (GMRFs). Hence, the joint posterior distribution of ϕ and θ is given by the product of the likelihood of Equation (10), of the GMRF density of Equation (11), and of the hyperparameter prior distribution of π(θ) given the data y, expressed as follows:π(ϕ,θ|y)∝π(θ)×π(ϕ|θ)×π(y|ϕ,θ)
$$\propto \pi \left(\theta \right)\times {\left|B\left(\theta \right)\right|}^{\frac{1}{2}}\exp\left(-\frac{1}{2}\phi^{T}B\left(\theta \right)\phi +\sum_{i=1}^{n}\log\left(\pi \left(y_{i}|\phi_{i},\theta \right)\right)\right)$$
$$\pi \left(\phi_{i}|y\right)=\int \pi \left(\phi_{i},\theta |y\right)d\theta$$
such that the individual elements of the hyperparameter vector $\pi \left(\theta_{k}|y\right)=\int \pi \left(\theta |y\right)d\theta_{-k}$. Therefore, through INLA, the calculation of the joint posterior of the hyperparameters via nested approximations is:$$\pi \left(\theta |y\right)=\frac{\pi \left(y|\phi ,\theta \right)\pi \left(\phi |\theta \right)\pi \left(\theta \right)}{\bar{\pi }\left(\phi |\theta ,y\right)}{|}_{\phi \theta =*\left(\phi \right)}$$

There are some mathematical functions that look complicated at first sight and nothing of significance can be drawn from them. Having such a function, it is good to investigate its local behaviour; thus, we gain an insight as to how the behaviour of the function itself will be. One of the tools in mathematics and statistics to do this is the Taylor series. In other words, the Taylor series is a mathematical tool used in approximating a function at a point excluding zero. If this point is zero, the Taylor series becomes the Maclaurin series. In the current study, the Laplace approximation technique constructed on the development of the Taylor series is used to approximate the posterior marginal π(ϕ|y) as follows:$$\bar{\pi }\left(\theta |y\right)=\frac{\pi \left(\phi ,\theta |y\right)}{\bar{\pi }\left(\phi_{-1i}|\phi_{1i},\theta ,y\right)}{|}_{\phi_{-i1=\phi_{-i1}^{*}\left(\phi_{i1},\theta \right)}}$$

In this case, $\bar{\pi }\left(\phi_{-1i}|\phi_{1i},\theta ,y\right)$ represent the Laplace gaussian approximation to $\pi \left(\phi_{-1i}|\phi_{1i},\theta ,y\right)$ and used to establish the nested approximations as $\bar{\pi }\left(\theta |y\right)\approx \int \bar{\pi }\left(\phi ,\theta |y\right)\bar{\pi }\left(\theta |y\right)d\theta$. The $\phi_{-1}\left(\phi_{i,}\theta \right)$ denote its node. Data were analyzed using the three-model specification above, namely the GAM, structured and unstructured model. Model diagnostic and comparison was based on deviance information criterion (DIC). The model regarded as the best is the one with the lowest DIC value. The model comparison statistics for all the models examined were presented in Appendix A. Therefore, final discussion of results was based on the third model (unstructured) which was the best fitted model with lowest DIC. Similarly, to account for the relative allowance of random effects in the model we employed the spatial variance ratio, which appraise the percentage of all random variance, a priori to the spatial random consequence.

## 3. Results

### 3.1. Sample Characteristics

Appendix A shows characteristics of the participants (N = 7824). Of these, 64.5% (n = 5893) were women, 83.8% (n = 6479) were never married, 57.6% (n = 4366) had incomplete high school education, 66.8% (n = 5088) had always lived in the community, the majority, 90.7% (n = 7110), had not been away from home for more than a month, 61.1% (n = 2917) had ran out of money in the last one year, and 65.1% (n = 2598) had a meal cut in the last year. Slightly more than half, (n = 4395) 58.3%, were accessing health care, 63% (n = 5039) of the participants had a monthly household income of ≤R2 500, while 10.4% (n = 939) were without an income. Just over half, 53.2% (n = 5222), were living in the peri-urban area. A majority, 84.4% (n = 6306), had reported being sexually active in the last year and 96.5% (n = 7257) responded that they were not forced the first time they had sex. Over a quarter (27.8%) (n = 1928) had ever consumed alcohol. Almost half (49.7% (n = 3990) had CD4 cell counts of 500 cells/µL and above. Towards achieving the UNAIDS 95-95-95 targets, 66.2% (n = 5191/7824) of HIV-positive individuals knew their status; 76.7% (n = 3937/5191) of HIV-positive individuals who knew their status self-reported being on ART and 84.9% (n = 3322/3937) of HIV-positive individuals who knew their status self-reported being on ART and had HIV viral load < 400 copies/mL. Thus, overall, 53.9% (n = 4259) of all PLHIV in this community had achieved viral suppression. (Appendix A).

### 3.2. Prevalence of Unsuppressed HIV Viral Load by Study Characteristics

Table 1 shows prevalence of unsuppressed HIV viral load by study sample characteristics. The overall prevalence was 46.1% (95% CI: 44.3–47.8; n/N = 3565/7824) and decreased from 50.9% (95% CI 47.3–52.7; n/N = 1981/3956) in 2014 to 42.0% (95% CI: 40.0–44.0; n/N = 1585/3868) in 2015 (*p*-value < 0.0001). Median age and interquartile range (IQR) were 30 and 25–46, respectively. Unsuppressed HIV viral load prevalence was higher among men, 54.2% (95% CI: 51.2–57.2), compared to women, 41.6% (95% CI: 39.8–43.4) (*p*-value < 0.0001). Stratified by age groups, prevalence of HIV viral load decreased as age increased, and it was 62.1% (95% CI: 55.4–68.8) among participants aged 15–19 years, 69.5% (95% CI: 65.5–73.5) in the 20–24 age group, 55.5% (95% CI: 52.3–58.8) in the 25–29 age group, 46.5% (95% CI: 43.4–49.6) in the 30–34 age group, 36.7% (95% CI: 33.2–40.2) in the 35–39 age group, 35.1% (95% CI; 31.7–38.6) in the 40–44 age group, and 30.2% (95% CI: 26.0–34.3) in the 45–49 age group, (*p*-value < 0.0001).

Prevalence of unsuppressed HIV viral load was slightly lower among those with no education, 46.3% (95% CI: 37.2–53.3), and those with incomplete high schooling, 42.5% (95% CI: 41.1–44.7), compared to those with complete high schooling, 48.4% (95% CI: 46.1–50.7) (*p*-value < 0.005). Prevalence was also higher among those that were never married, 47.1% (95% CI: 45.6–48.7), compared to those married, 34.4% (95% CI 32.0–38.9) (*p*-value ≤ 0.0001). Higher prevalence of unsuppressed HIV viral load was observed among those who had been away from home for more than a month, 54.2% (95% CI: 49.2–59.2), compared to those that were not away from home, 45.2% (95%: 43.4–47.0) (*p*-value = 0.0013). Slightly higher prevalence was found among those residing in the peri-urban area, 45.9% (95% CI: 44.1–47.6), compared to those in the rural area, 43.9% (95% CI: 41.4–46.4) (*p*-value = 0.2057). Those who reported “no monthly household income” had a higher prevalence, 49.9% (95% CI 46.1–53.7), compared to those who reported household monthly income earning of R2500 or less, 46.9% (95% CI: 44.7–49.0) (*p*-value = 0.0042).

### 3.3. Prevalence by Behavioural Factors, Perception, and Knowledge of HIV Testing Variables

Those who reported “ever consumed alcohol” had a higher prevalence of unsuppressed HIV viral load, 56.4% (95% CI: 53.4–59.4), compared to those who never consumed alcohol, 42.1% (95% CI: 40.2–44.0) (*p*-value < 0.0001). Unsuppressed HIV viral load prevalence was slightly higher among participants who reported that they did not remember being “forced first time they had sex”, 59.4% (95% CI: 46.6–71.9), and those reported that they were not forced the first time they had sex, 46.0% (95% CI: 44.2–47.5), compared to those that reported they were forced first time they had sex, 41.1% (95% CI: 34.5–49.5) (*p*-value = 0.0004). A higher prevalence of unsuppressed HIV viral load was observed among those that had never tested for HIV, 76.2% (95% CI: 72.2–80.3), than those that had tested for HIV, 42.6% (95% CI: 40.8–44.3) (*p*-value < 0.0001).

Participants who reported “perceived risk of not likely to acquire HIV” were observed to have slightly higher unsuppressed HIV viral load prevalence, 74.6% (95% CI: 72.0–77.2), compared to those that perceived they were likely to acquire HIV, 73.1% (95% CI: 69.9–76.2) (*p*-value < 0.0001).

### 3.4. Prevalence by History of Tuberculosis, Sexually Transmitted Infections, and Clinical Characteristics

Prevalence of unsuppressed HIV viral load was significantly lower among those ever tested for tuberculosis (TB), 32.3% (95% CI: 30.2–34.4) (*p*-value < 0.0001), among those exposed to TB in the last 12 months, 39.5% (95% CI: 33.8–45.5) (*p*-value = 0.003), and among those on TB medication 20.8% (95% CI: 17.3–31.7) (*p*-value < 0.0001), compared to those that never tested, were never exposed to TB, or were not on TB medication. Furthermore, prevalence of unsuppressed HIV viral load was slightly lower among those who reported they ever had any STI symptoms, 43.8% (95% CI: 36.8–50.8), compared to those that never had any symptoms, 46.2% (95% CI: 44.4–47.9) (*p*-value = 0.52). Higher prevalence was observed among those who reported that they had been diagnosed with an STI, 51.5% (95% CI: 46.7–56.3), compared to those that were never diagnosed with an STI, 45.4% (95% CI: 43.5–47.3) (*p*-value = 0.02).

Prevalence of unsuppressed HIV viral load decreased as CD4 cell counts increased, and among those with CD4 cell counts of <350 cell/µL, prevalence was 68.4% (95% CI: 65.6–71.2), 47.1% (95% CI: 43.9–50.3) among those between 350–499 cell/µL, and 32.9% (95% CI: 30.7–33.0) among those with ≥500 cell/µL (*p*-value < 0.0001).

### 3.5. Spatial Variation in Unsuppressed HIV Viral Load Prevalence

In examining the spatial pattern of unsuppressed viral load in this hyperendemic enumeration area, Geary’s C-test for spatial autocorrelation showed significant evidence of spatial variation: [observed (O) = 1.0019, expected (E) = 1.0000, standard deviation (SD) = 0.000057, Z score = 3.34, and *p* < 0.0008]. This showed that unsuppressed HIV viral load was randomly distributed in the area. Figure 2 shows the observed prevalence map of unsuppressed viral load; two localized high-prevalence areas were identified; these were in the north (rural) and northeast (peri-urban) of the study area. Furthermore, Mpophomeni, Edendale, Imbali, Ashdown, and a part of Pietermaritzburg (light and bright yellow colour) were identified as areas with higher prevalence of unsuppressed viral load compared to other parts of the study area.

Initial non-spatial bivariate survey logistic regression analysis was implemented to test for each independent association with the dependent variable. All variables were independently associated except enumeration area, ever had sex, and ever had STI symptoms (Appendix A). All associated independent variables that were statistically significant at a 5% level of significance were then considered in the multivariable spatial model. Continuous variables included in the multivariable analysis which were modelled non-parametrically were current age, size of household, and number of lifetime HIV tests.

Appendix A shows values of the DIC used in determining the best-fitted model. The model with the smallest DIC value is considered best-fit and is preferred. From the findings, the unstructured model has the minimum value (DIC = 5894.51), thus attesting as the best-fit model for the data sets, while GAM model offers the least fit. In addition, the unstructured model is of actual interest because it contains all of the variables considered, and accounts for spatial variation and between-cluster heterogeneity.

Findings for the posterior mean and 95% credibility interval for the smooth-term variance components for both non-linear and spatial effects are presented in Table 2. The accuracy of an effect is the inverse of its variance. Consequently, the higher the accuracy, the smaller the variance of the effect. The precision corresponding to the structured spatial effect (87.97) was much higher than that of the unstructured spatial effect (27.934), thus suggesting that the unstructured spatial effect was more assertive, and previous studies support this observation [63,75,76].

### 3.6. Predictors of Unsuppressed HIV Viral Load among HIV-Positive Men and Women in Rural and Peri-Urban Areas of KZN

Table 3 shows the estimates for the Bayesian geoadditive model based on the posterior-adjusted odds ratio (AOR) and 95% CrI. The findings showed that the study year was a predictor of unsuppressed HIV viral load. The results also revealed that participants with incomplete high school education (AOR = 0.019; 95% CrI: 0.002–0.035) have higher odds of being unsuppressed compared with those with complete high school education. We found that not being away from home for more than one month reduced the risk of being virally unsuppressed (AOR = 0.043; 95% CrI: 0.016–0.071), compared to those who were away from home. Similarly, those who ever consumed alcohol (AOR = 0.057; 95% CrI: 0.037–0.077) were more likely to be at risk of being unsuppressed than those that never consumed alcohol. No prior knowledge of HIV status was significantly associated with a higher risk of unsuppressed HIV viral load (AOR = 0.158; 95% CrI: 0.124–0.193). Results based on ever tested for HIV reveal that participants who had never tested for HIV were more likely to be virally unsuppressed (AOR = 0.050; 95% CrI: 0.031–0.069). Additionally, there was a higher likelihood of being virally unsuppressed among those not currently on ART (AOR = 0.377; 95% CrI: 0.342–0.412) compared with those on ART, and for those on ART, being on multiple ARVs (AOR = 0.204; 95% CrI: 0.178–0.239), rather than being on a fixed-dose combination regimen, was associated with a higher risk. We also found that those that ever tested for TB (AOR = 0.050; 95% CrI: 0.031–0.069) and those on TB medication (AOR = 0.027; 95% CrI: 0.001–0.055) were more likely to be at risk of virally unsuppressed compared to their reference categories.

Considering individual perceived risk of contracting HIV, those with moderate to low perceived risk of contracting HIV had a higher risk of being virally unsuppressed compared to those that already knew they had been infected with HIV. Further, having two or more sex partner in the last 12 months (AOR = 0.027; 95% CrI: 0.005–0.049) increased the odds of being unsuppressed. Lastly, we found a lower likelihood of being unsuppressed with current CD4 cell counts between 350–499 cells/µL (AOR = 0.161; 95% CrI: 0.138–0.183) and CD4 cell counts ≥ 500 cells/µL (AOR = 0.270; 95% CrI: 0.251–0.289), compared with those in the category of < 350 cells/µL.

### 3.7. Non-Linear Effect of Covariates and Spatial Effects Map

Figure 3 shows the non-linear effects of current age in years, household size, and number of lifetime HIV tests and corresponding 95% confidence intervals (CIs) represented by the shaded region from the best-fitted model.

Figure 3a shows that the non-linear effect of age on the log-odds of unsuppressed viral load is inversely related to age. Additionally, the figure shows that the effect of age on the log-odds of unsuppressed viral load decreases as age increases; however, those between ages 15 to 25 years old have a higher risk of being virally unsuppressed as indicated by the positive log-odds and a U-shaped curve. Figure 3b shows that the effect of household size on the log-odds of unsuppressed viral load increases as household size increases. The plot indicates that the effect of household size on unsuppressed viral load was almost constant and was only significant at household sizes of about 5 and 10. Figure 3c shows that the effect of number of lifetime HIV tests on the log-odds of unsuppressed viral load slightly increases as the number of lifetime HIV tests increases. From the plot, the effect was mostly constant and only shows slight significance at lifetime HIV test count of 30 above, indicating that risk of unsuppressed viral load increases with a higher number of lifetime HIV tests.

Figure 4 shows the choropleth spatial effect maps for the structured and unstructured model. Both maps revealed positive and negative effects on unsuppressed HIV viral load among HIV-positive men and women. The colours on the choropleth map show the log-odds scale, indicating the influence of enumeration area and its contribution to the odds of individual men and women with unsuppressed HIV viral load. The choropleth map also reveals that enumeration areas with dark blue, orange, and yellow colours have significant positive spatial effect of posterior mean between 4 to 2 (structured map) (Figure 4a) and 0.1 to 0.2 (unstructured map) (Figure 4b) and are associated with higher odds of unsuppressed HIV viral load. Royal to dark blue shaded areas show negative spatial effect between −4 to −2 (structured map) and between −0.1 to −0.2 (unstructured map) and are associated with significantly lower odds of being virally unsuppressed. The pink and purple colours show areas with non-significant effect. There exists a strong indication of spatial variation between men and women who were virally unsuppressed in this community.

## 4. Discussion

This study showed the prevalence, spatial variation, and predictors of unsuppressed HIV viral load among HIV-positive men and women at a population level whilst accounting for unobserved factors. Using advanced statistical methods of a Bayesian framework, the study identified the non-linear effect of selected continuous variables and mapped the spatial effects to delineate the high-risk areas in an HIV-hyperendemic region in the Vulindlela and Greater Edendale areas of uMgungundlovu District in KwaZulu-Natal Province, South Africa. Our analysis draws on the data from the HIPSS sequential cross-sectional survey data and is based on a sample of 7824 HIV-positive participants with viral load measurements. We found significant spatial variation of unsuppressed HIV viral load among men and women across the enumeration areas.

A Bayesian geoadditive model, which includes a spatial effect for a small enumeration area along with semi-parametric unstructured and structured geoadditive regression models, was applied via an integrated nested Laplace approximation (INLA) function while controlling for the confounding effects of the explanatory variables using HIPSS data. Bayesian inference was applied using INLA, due to its numerous advantages over classical statistics [67,68,69,70]. The Bayesian spatial approach is beneficial in minimizing bias and variance compared to classical statistical approaches. Furthermore, it is superior when models contain residual spatial dependence because the approach offers robust and more consistent estimates via capturing the uncertainty that is intricate in parameter estimation [73,75].

Taking into consideration the residual uncertainty in our model, especially when faced with computational complexity due to potential space variation and dependent relationship between predictors and response variables, the method helps increases prediction accuracy, with ability to produce smooth risk maps and increase precision by accounting for spatial dependencies in the model [76]. Due to the strength of this approach, many studies have investigated risk factors of anaemia and other infectious diseases that have emanated across sub-Saharan Africa [77,78,79,80], of HIV variation in Ethiopia [81], Kenya [82], Nigeria [83], and Oman in Western Asia [40]. A previous study from KZN among women used multilevel logistic regression and Kulldorff statistics to identify high-risk areas [29].

Given the complexity of our dataset and that of small EAs, the Bayesian spatial approach was highly desirable in producing reliable and accurate results, as it accounted for unobserved factors in this location while assigning prior knowledge to relax any form of uncertainty in determining predictors of unsuppressed viral load in this community. Thus, the approach highlighted the specific areas that require urgent attention in order to facilitate access to HIV testing with better linkage to care and ART initiation.

Our findings revealed that the year of the study, incomplete high school education, being away from home for more than a month, ever consumed alcohol, no prior knowledge of their HIV status, not ever tested for HIV, presently not on antiretroviral therapy (ART), ever tested for TB, on tuberculosis (TB) medication, having two or more sexual partners in the last 12 months, having a perceived risk of contracting HIV, and having a CD4 cell count of <350 cells per μL were significant predictors of unsuppressed HIV viral load among men and women aged 15–49 years old.

Thus, the present study further highlights the urgent need of ensuring ART initiation and sustainability toward achieving viral suppression for individual and population benefit. Following the substantial benefit in the use of ART [26,84], dosage of ART drug levels was found to be significantly associated with being virally suppressed, which is similar to past studies [41,85,86,87]. Therefore, to improve adherence, the first-line ART regimen has been simplified and improved to a single tablet of fixed-dosed combination (consisting of tenofovir, emtricitabine, and efavirenz) [85,87].

Alcohol consumption was found to be significantly associated with unsuppressed viral load. This was similar to the findings among younger women living with HIV in South Africa [88] and HIV patients on first-line ART therapy in Morocco [89], but in contrast with studies from the USA, where alcohol consumption was not a significant predictor [90]. South Africa is known for its high volume of per capita alcohol consumption amongst people who drink, while a substantial proportion (>50%) of the adult population abstain from consuming alcohol, heavy episodic drinking (HED) is the normative pattern of alcohol use amongst people who consume alcohol [88]. Previous studies have also shown the effect of alcohol consumption on the immunological responses by increasing inflammatory responses that potentially worsen disease progression [91,92]; it also affects the metabolism and effectiveness of antiretroviral drugs [93]. Alcohol use has been found to be associated with poor long-term adherence to treatment in PLHIV [93,94], the prevalence of which leads to high risk of HIV transmission [95]. Intervention targeting reduction in alcohol consumption may help the community augment their adherence to ART and potentially lower transmission of HIV.

Having ever been tested for TB and being on TB medication were found to be predictors of unsuppressed viral load in this population. Several cross-sectional studies have emphasized the relationship between TB occurrence and unsuppressed HIV viral load [86,87,88]; however, the causal relationship may not be clear. Our findings were consistent with studies from Uganda [96] and South Africa [97,98], with 75% and 33% prevalence of unsuppressed HIV viral load among PLHIV with TB comorbidity. Similarly, the risk of unsuppressed HIV viral load may also be increased due to impaired treatment adherence and pharmacokinetics drug interaction with concurrent ART and TB treatment [41] and, therefore, these individuals should be prioritized for viral load monitoring and adherence support interventions.

Furthermore, our study findings showed that individual current CD4 cell count at diagnosis was also a predictor and PLHIV, with higher CD4 cell counts of between 350–499 and ≥500 cells/µL more likely to be associated with viral suppression. This was similar to findings in Uganda among female sex workers on ART [99] and also a study in Nigeria among men and women, which showed higher CD4 cell counts ≥500 cells/µL to be predictors against opportunistic diseases and viral suppression [100]. In contrast, studies from Ethiopia and Guatemala found PLHIV with lower CD4 cell counts >200 cells/µL was more likely to be virally unsuppressed [85,101]. A study has shown that there exist interactions between CD4 cell count, viral suppression, and ART treatment [102]. CD4 cells are one of the prime targets of HIV; hence, its fall corresponds to increasing viral load. The findings of low-baseline CD4 cell count as predictive of unsuppressed HIV viral load affirms the previous observation that individuals with high viral load tend to have low CD4 cell count, resulting in slow viral suppression and clearance [103]. A higher proportion of men and women with unsuppressed viral load (68.4%), and with a lower CD4 cell count, >350 cells/µL, suggests that they were not on ART and might not be aware of their HIV status, even with marked immunosuppression. CD4 cell count is used for early prediction of virological failure during antiretroviral treatment in a resource-limited setting [104]. Further, the World Health Organization (WHO) has recommended the use of a CD4 cell-count measure for monitoring ART in the absence of viral load, which several studies have proven to be effective [104,105]. Therefore, the body is vulnerable to opportunistic infections when the CD4 cell count is below 200 or 350 cells/µL [106].

Socio-demographic factors, such as incomplete high school education and migration history of being away from home for more than one month and behavioural variables such as having two or more sexual partners, were also found to be predictors of unsuppressed viral load in this community. These are similar to findings across sub-Saharan Africa [29,107] and in other developed countries [108]. Higher education level reduces the risk of HIV infection and positively impacts sexual behaviours; young people with higher and longer stays in school are more likely to be aware of HIV and AIDS, know their status, be more inclined to the uptake of ART, and take protective measures by using condoms and discussing with their sexual partners. Having a low perceived risk of contracting HIV was also found to be predictor of unsuppressed viral load in this community compared to those that already knew they had contracted the virus. A recent study in Guatemala revealed that negative perception is a barrier to accessing necessary HIV care service which, in turn, is associated with virological failure [101].

The overall unsuppressed viral load prevalence of 46.1% in this study was higher compared to a study among adults in Ethiopia, reporting a prevalence of 26.39% [85], in Malawi, among adults, with a prevalence of 39% [109], and Uganda, with a prevalence of 11% [41]. Conversely, whilst 53.9% of participants were virally suppressed at HIV viral load of <400 copies/mL, the proportion reported was far below the 72% reported for the province of KwaZulu-Natal [3] and 65% reported nationally for South Africa at the end of 2020 [2]. The highest prevalence of unsuppressed viral load was observed among younger age group 15 to 29 years old, with the lowest prevalence observed among the 40–49-year-old age group. It is worth nothing that an increase in the HIV viral suppression of 8.9% was observed in this community over a one-year period from the baseline survey (2014 survey) to the 2015 survey, and this could be attributed to the country-wide adoption of UTT to compliment case-findings strategies as reflected in the revised 2016 National HIV Testing Services (HTS) policy [28,110]. However, the lag time from policy to implementation could result in substantial delays and affect the roll-out and uptake of services, specifically the uptake of ART, resulting in the high prevalence of unsuppressed HIV viral load.

Additionally, we found that current age, household size, and number of lifetime tests had a non-linear effect on risk of being virally unsuppressed among men and women in this community. The log-odds of having unsuppressed viral load decreases with age, increases with a higher number of household size, and increases with a higher number of lifetime HIV tests. Although no study has shown the non-linear effects of continuous covariates on risk of unsuppressed viral load, conventional approaches showed that age has been found to be a predictor, with younger age groups having higher risk of being virally unsuppressed compared to older age groups [41,96,107]. This could be attributed to adolescent pressure, non-adherence, and stigma.

Furthermore, spatial effect maps show substantial high spatial variation in the north region of the study area, indicating evidence of observed spatial heterogeneity on predictors of unsuppressed viral load among HIV-positive men and women. Not accounting for spatial variability could result in skewed results, biased estimates, inappropriate decisions, and conclusions which past studies did not implement [111]. The geographic variation in infectious diseases such as HIV could be one cause of such spatial variation as we observed in this study. Areas with higher, lower, and moderate risks of unsuppressed viral load in relationship to their neighbours were detected. The unstructured spatial effect map, which is of utmost interest, significantly showed northeast and northwest regions as areas with predicted higher risk of unsuppressed viral load. These areas are cosmopolitan peri-urban locations explained by the characteristics of urbanization and diversity [5]. This study also revealed that the structured spatially correlated effects were not significant, while the unstructured spatial correlated effects was found to be significant on viral load status. Based on these findings, it can then be concluded that there are unmeasured location-specific factors contributing to unsuppressed viral load. The information on spatial heterogeneity of this area will critically aid epidemiologists in optimising engagement in care and service delivery. Additionally, identifying high-risk locations will enhance targeted resources and location-based programmes and intervention to increase viral suppression in this area. Spatial effect incorporated into the model helped to avoid underestimating the standard errors of the estimates of the model parameters, which many past studies investigating factors associated with unsuppressed viral load did not consider [29,30,32,43].

Additionally, UNAIDS 95–95-95 targets had not been met in this study area. Of those PLHIV, 66.2% were aware of their HIV status; 76.7% of these were on ART, and of those knowledgeable of their status and on ART, 84.9% had suppressed viral load, with a composite viral suppression of 53.9% (Appendix A). The composite viral suppression of 53.9% among HIV-infected men and women in this hyperendemic area of KZN is substantially below the UNAIDS target of 86% to achieve the goal of epidemic control by 2030, and of the country, with 85.6% at viral suppression threshold of <400 copies/mL at the end of 2020 [4]. The implication of this difference between the UNAIDS 1000 copies/mL and that of our study and that of the country, 400 copies/mL cut-off for viral suppression, is the challenge of adherence to ART in South Africa, despite having the largest ART program globally, therefore suggesting targeted resources to improve retention and adherence. More importantly, determining factors associated with each of the UNAIDS indicators using advanced statistical approach is recommended. Furthermore, in this hyperendemic region, our findings shows that viral suppression of 58.4% among women and 45.8% among men was encouraging, though far short of the UNAIDS target toward achieving control of the epidemic. These was also lower than findings among women in Zambia (61.3%, 95% CI: 58.7–63.8), Zimbabwe (64.5%, 95% CI: 62.2–66.7), and Malawi (72.9%, 95% CI: 69.9–75.9), all having higher viral suppression [112]. Similarly, a recent study among pregnant adolescent and women living with HIV in the rural area of Vulindlela, KZN, showed 40.2% composite viral suppression [113]. Thus, our findings among women in this community suggested that the health systems are in place for HIV care trajectory and that women were more likely to adhere to ART in contrast to men. Therefore, a targeted approach to increase the uptake of HIV testing and linkage to immediate treatment to achieve and sustain HIV viral suppression, specifically for men, are urgently needed. Although gender was not a significant factor in this study, based on the differences in the structural and behavioural characteristics of men and women, as well as higher prevalence of unsuppressed viral load among men in this study, further analysis on gender specificity is recommended. Specifically, this will further provide more targeted information for epidemiologists on predictors of unsuppressed viral load among men and women.

This study contributed to existing knowledge by applying existing methodology, namely a Bayesian geoadditive model in R-INLA, to examine predictors and high-risk areas of unsuppressed viral load among men and women in a small enumeration area. To our knowledge, this study is the first to use Bayesian geoadditive modelling to capture the complexity of HIPSS data by incorporating the spatiotemporal analysis of the study years and achieving the study set objectives at a population level with considerable large sample size, which provide a meaningful analysis and generalizability. A previous study from the same area and data have examined only women without incorporating spatiotemporal analysis in the model [114]. This population-level study with robust data on socio-demographic, behavioural, and psycho-social factors, knowledge of HIV testing and testing history, TB and STI history, and clinical and spatial characteristics in this hyperendemic area will contribute to the knowledge of understanding holistic predictors of unsuppressed viral load as well as identifying areas to be targeted for rapid intervention and implementation of programmes and resources that will help in achieving detectable viremia among PLHIV. Other key strengths of our study were the robustness of the study design, large sample size (which enabled generalizability), high participation rates, availability of spatial variables, and the conducting of the survey in a real-time setting. Therefore, we assume that the findings of this present work will be used by epidemiologists, public health institutes, and policy makers in their efforts to meet the UNAIDS target, in intensifying prevention and transmission measures, and in the uptake of treatment and programmes. It will also assist in focusing on this identified high-risk area to enhance planning to increase viral suppression among PLHIV.

However, the findings should be interpreted with the following limitations in view: Viral load testing was based on a single point of testing after diagnosis of being HIV-positive; causality or timing of virological non-suppression was not taken into consideration. The study is a cross-sectional and population-based surveillance, rather than a randomized clinical trial of monitoring initiation of ART treatment. We were not justified in undertaking a randomized control trial; instead, we aimed to examine whether there are impacts of the health sector’s programmes on HIV outcomes. As recommended by one of the reviewers, further research and analysis focusing on the type of enumeration area (peri-urban vs. rural) to seek factor variation according to enumeration area type is, therefore, recommended. This will aid in developing geographic location-transformative HIV policies, strategies, and programmes that will aid in reaching key population goals and achieve the epidemic targets by 2030.

## 5. Conclusions

This study was motivated by the need to identify predicted high-risk locations and the predictors of unsuppressed viral load in this HIV-hyperendemic community. The HIV burden in this small enumeration area with suboptimal differential coverage and uptake of HIV prevention strategies justify a location-based approach to surveillance and spatial modelling. Considering the revised UNAIDS 95-95-95 and 86% composite viral suppression targets to be attained by 2030, it is vitally important to exponentially scale-up effective targeted risk-based HIV prevention strategies and holistically double the efforts to combat the spread of this infectious disease in South Africa.

Bayesian geoadditive modelling was applied on HIPSS data based on the Monte Carlo Markov Chain (MCMC), which is a powerful approach and has gained more interest due to its strength compared to other conventional statistics modelling approaches. We hypothesized that the odds of having unsuppressed viral load is not only associated with clinical variables such as CD4 cell count and ART uptake, but is also influenced by socio-economic (year of study, education level, being away from home for more than 1 month) and behavioural factors (alcohol consumption, number of sex partners last 12 months), medical history (ever tested for TB and on medication to prevent TB), and knowledge of HIV testing and perceived risk (knowledge of HIV status, ever tested for HIV, and perceived risk of contracting HIV) variables. Intervention strategies that could ensure behavioural factors and HIV testing knowledge and perception of HIV risk need to be adopted for successful control, prevention, attainment of undetectable viral load among PLHIV.

Considerable spatial variation exists across this small enumeration area, which this study accounted for in identifying predictors of unsuppressed viral load. The spatial analysis method allows for identification of locations with significantly higher or lower risk. From our study, predicted high-risk areas of unsuppressed viral load were identified as the northeast and northwest areas, while accounting for covariate measure. This provides insight into the effective allocation of resources through cost-effective and efficient HIV prevention strategies by prioritization of needs in high-risk areas and key populations. Along with looking at regional and country-specific strategies in tackling this epidemic, focus should also be on small areas and district-specific strategies.

The overall prevalence of unsuppressed viral load of 46.1% is high among men and women in this HIV-hyperendemic community, with men having a higher prevalence of unsuppressed viral load of 54.2% compared to women. The composite viral suppression of 53.9% in this study area is substantially below the UNAIDS targets of 85% to end the epidemic by 2030. However, an improvement by 8.9% over the study period was observed. This shows that the target is achievable in this localized region, which also reflects South Africa’s commitment in scaling-up ART uptake and increasing efforts in HIV prevention programmes towards ending the HIV epidemic by 2030.

Evidently, there exists a positive non-linear association between current age, household size, and number of lifetime HIV tests on the log-odds of unsuppressed viral load among men and women in this HIV-hyperendemic area. Thus, in modelling this consequence, pretentious linear covariates would have led to spurious findings and a less penurious model. Additionally, this would have resulted in erroneous inferences, which could misinform policymakers. Adolescent and gender-specific interventions such as improving ART adherence and the reduction of stigmatisation are recommended, thereby translating political commitment into programmatic actions.

Additionally, seeing the first 95 and the second 95 indicators as predictors of unsuppressed viral load, assessing the determinants of each of the 95-95-95 targets is hereby recommended to further inform public health institutions and epidemiologists, as well as to aid in fast-tracking the UNAIDS goal. While many countries are striving to achieve the UNAIDS 95-95-95 targets and control the epidemic, PLHIV in a hyperendemic setting, where the rate of viral suppression is low, need holistic targeted programmes that will create more awareness and increase their knowledge.

Accounting for spatial heterogeneity and time while identifying predictors of unsuppressed viral load in a localized geographic area and information from risk effect maps are highly relevant in substantially reducing high transmission of HIV, increasing aid-effective targeted prevention and treatment programs, and helping to fast-track the UNAIDS goal in this HIV-hyperendemic area.

## Figures and Tables

**Figure 1 tropicalmed-07-00232-f001:**
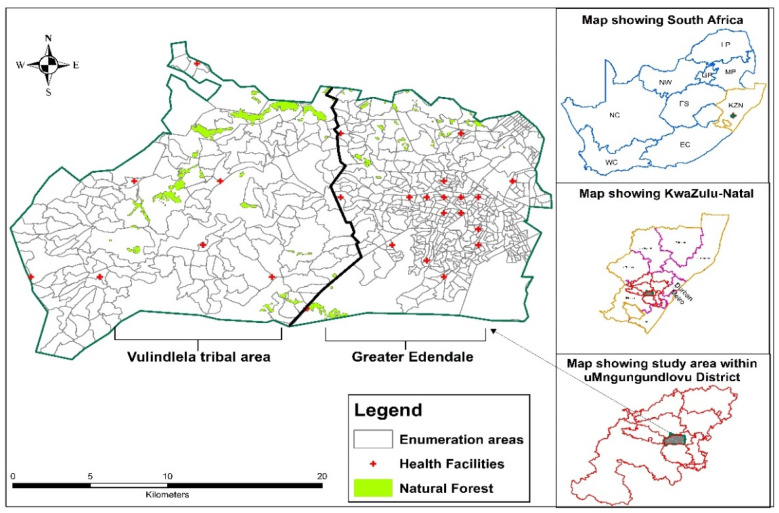
Location map of Vulindlela (rural) and Greater Edendale (peri-urban) area in the uMgungundlovu District, KwaZulu-Natal Province, South Africa.

**Figure 2 tropicalmed-07-00232-f002:**
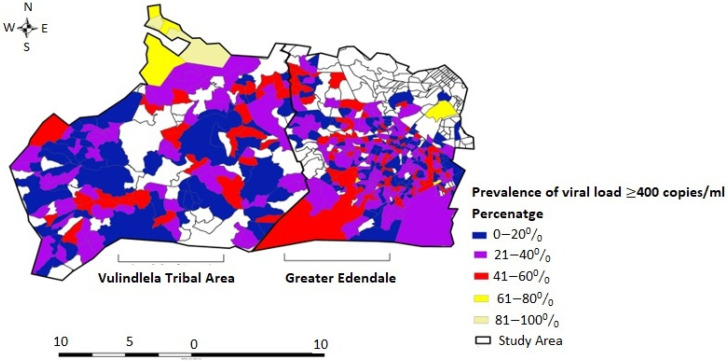
Observed prevalence map of unsuppressed HIV viral load among HIV-positive men and women in rural Vulindlela and peri-urban Greater Edendale area in the uMgungundlovu District, KwaZulu-Natal Province, South Africa. Geographic coordinate scale at five-kilometre radius.

**Figure 3 tropicalmed-07-00232-f003:**
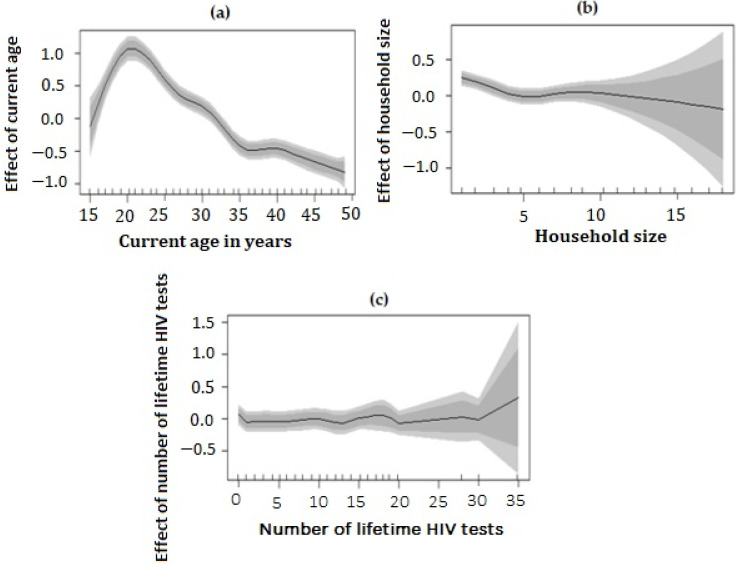
Estimated non-linear effects of (**a**) current Age (in years), (**b**) household size, and (**c**) number of lifetime HIV tests on the log-odds of unsuppressed viral load among HIV-positive men and women in Vulindlela (rural) and Greater Edendale (peri-urban) areas in uMgungundlovu District, KwaZulu-Natal Province, South Africa. The posterior means and 95% credible intervals are also shown by the shaded region.

**Figure 4 tropicalmed-07-00232-f004:**
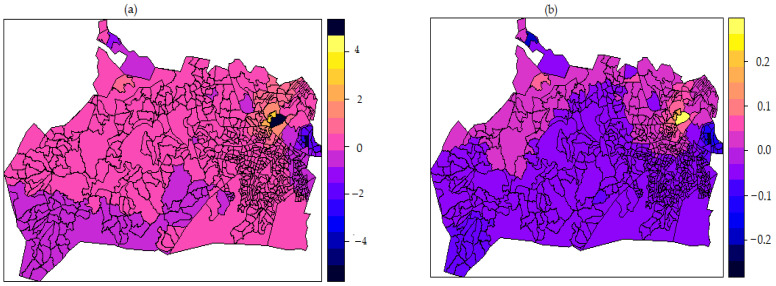
Map of estimated posterior mean for (**a**) structured and (**b**) unstructured spatial effect on the log-odds of unsuppressed HIV viral load among HIV-positive men and women in rural Vulindlela and peri-urban Greater Edendale areas in uMgungundlovu District, KwaZulu-Natal Province, South Africa.

**Table 1 tropicalmed-07-00232-t001:** Prevalence of unsuppressed HIV viral load by participant characteristics among HIV-positive men and women in Vulindlela (rural) and Greater Edendale (peri-urban) areas in the uMgungundlovu District, KwaZulu-Natal Province, South Africa.

Characteristics	HIV Viral Load <400 Copies/mL		HIV Viral Load ≥400 Copies/mL		*p*-value
	(Suppressed)		(Unsuppressed)		
	n = 4259	53.9% (52.2–55.7)	n = 3565	46.1% (44.3–47.8)	
** *Socio-demographic* **					
Age (median, IQR)	35 (29–48)		30 (25–46)		
Household size (median, IQR)	3 (2–8)		3 (2–8)		
**Year**					
2014	1975	49.1% (48.7–52.7)	1981	50.9% (47.3–52.7)	<0.0001
2015	2284	58.0% (56.0–61.0)	1584	42.0% (40.0–44.0)	
**Gender**					
Male	857	45.8% (42.8–48.8)	1074	54.2% (51.2–57.2)	<0.0001
Female	3402	58.4% (56.6–60.1)	2491	41.6% (39.8–43.4)	
**Age group (in years)**					
15–19	126	37.9% (31.2–44.6)	211	62.1% (55.4–68.8)	<0.0001
20–24	300	33.1% (26.5–34.5)	629	69.5% (66.5–73.5)	
25–29	687	44.5% (41.2–47.7)	797	55.5% (52.3–58.8)	
30–34	910	53.5% (50.3–56.6)	759	46.5% (43.4–49.6)	
35–39	911	63.3% (59.8–66.8)	504	36.7% (33.2–40.2)	
40–44	789	64.9% (61.4–68.3)	422	35.1% (31.7–38.6)	
45–49	536	69.8% (65.7–73.9)	243	30.2% (26.0–34.3)	
**Education level ^A^**					
Incomplete High schooling	2473	57.5% (53.3–57.8)	1893	42.5% (41.1–44.7)	0.0005
Completed High schooling	1665	51.6% (49.3–53.9)	1566	48.4% (46.1–50.7)	
No Schooling	121	53.7% (44.7–62.8)	106	46.3% (37.2–55.3)	
**Away from home for >1 month ^B^**					
Yes	330	45.8% (40.8–50.8)	384	54.2% (49.2–59.2)	0.0013
No	3929	54.8% (53.0–56.6)	3181	45.2% (43.4–47.0)	
**Community duration ^C^**					
Always	2673	52.5% (50.5–54.6)	2415	47.1% (45.4–48.8)	<0.0001
Moved here less than 1 year ago	131	50.4% (42.7–58.1)	135	49.6% (41.9–57.3)	
Moved here more than 1 year ago	1455	57.3% (54.6–60.0)	1015	42.7% (40.0–45.4)	
**Marital Status**					
Never married	3395	51.9% (50.0–53.8)	3084	48.1% (46.2–50.0)	<0.0001
Married	864	64.2% (60.6–67.9)	481	35.8% (32.1–39.4)	
**Enumeration area**					
Rural	1461	56.1% (53.6–58.6)	1141	43.9% (41.4–46.4)	0.2057
Urban	2798	54.1% (52.4–55.9)	2424	45.9% (44.1–47.6)	
**Run out of money last 12 months ^D^**					
Yes	1646	54.8% (52.3–57.3)	1271	45.2% (42.6–47.7)	0.0650
No	2605	53.3% (51.9–55.41)	2288	46.7% (44.5–48.8)	
**Had meal cut last 12 months ^E^**					
Yes	1468	54.9% (52.1–57.6)	1130	45.1% (42.4–47.9)	<0.0001
No	2783	53.4% (51.4–55.4)	2429	46.7% (44.6–48.6)	
**Accessing health care ^F^**					
Yes	2646	58.5% (56.2–60.9)	1749	41.5% (39.1–43.8)	<0.0001
No	1605	47.5% (45.1–49.8)	1810	52.5% (50.2–54.9)	
**Monthly Income**					
No income	465	50.1% (46.2–53.9)	474	49.9% (46.1–53.7)	0.0042
≤ R2 500	2725	53.1% (51.0–55.3)	2314	46.9% (44.7–49.0)	
> R2 500	1063	57.3% (54.5–60.1)	773	42.7% (39.9–45.5)	
** *Behavioural* **					
**Number of sex partner last 12 months (median, IQR)**		1 (1–2)		1 (1–2)	
**Number of lifetime sex partners (median IQR)**		3 (2–5)		3 (1–5)	
**Had sex in last 12 months**					
Yes	3398	53.4% (51.5–55.3)	2908	46.6% (44.7–48.5)	0.14
No	731	56.2% (52.7–59.7)	492	43.8% (40.3–47.3)	
**Forced sex first time**					
Yes	110	58.9% (50.5–67.5)	72	41.1% (34.5–49.5)	0.0004
No	4106	54.0% (52.2–55.8)	3281	46.0% (44.2–47.8)	
Don’t remember	43	44.6% (39.5–49.9)	212	59.4% (46.6–71.9)	
**Alcohol consumption**					
Yes	829	43.6% (40.6–46.6)	1099	56.4% (53.4–59.4)	<0.0001
No	3430	57.9% (56.0–59.8)	2466	42.1% (40.2–44.0)	
** *HIV perception and testing knowledge* **					
**Number of lifetime HIV tests (median, IQR)**		2(1–4)		2(1–3)	
**Ever tested for HIV**					
Yes	4020	57.4% (55.7–59.2)	2939	42.6% (40.8–44.3)	<0.0001
No	239	27.8% (19.7–29.8)	626	76.2% (72.2–80.3)	
**Perceived risk of contracting HIV**					
Likely to Acquire HIV	448	26.9% (23.8–30.1)	1028	73.1% (69.9–76.2)	<0.0001
Not likely to Acquire HIV	429	25.4% (22.8–30.0)	1142	74.6% (72.0–77.2)	
Already infected	3382	72.2% (70.4–73.9)	1395	27.8% (26.1–29.6)	
** *Medical history* **					
**Ever tested for TB**					
Yes	2622	67.7% (65.6–69.8)	1240	32.3% (30.2–34.4)	<0.0001
No	1637	38.9% (36.9–41.0)	2325	61.1% (59.0–63.1)	
**Exposed to TB last 12 months**					
Yes	230	60.0% (56.6–69.4)	139	39.5% (33.8–45.5)	0.003
No	4022	53.5% (51.8–55.2)	3420	46.5% (44.8–48.2)	
**Ever diagnosed with TB**					
Yes	687	71.2% (67.3–75.2)	272	28.2% (24.9–31.7)	0.04
No	3572	51.1% (49.3–58.8)	3293	48.9% (47.2–50.7)	
**On TB medication**					
Yes	629	79.2% (75.8–82.7)	179	20.8% (17.3–24.2)	<0.0001
No	3630	51.0% (49.2–50.8)	3386	49.0% (47.2–50.8)	
**Ever had any STI symptoms**					
Yes	174	56.2% (49.3–63.2)	132	43.8% (36.8–50.8)	0.52
No	4085	53.8% (50.1–55.6)	3433	46.2% (44.4–47.9)	
**Ever diagnosed with STI**					
Yes	373	48.5% (43.7–53.3)	370	51.5% (46.7–56.3)	0.02
No	3886	54.6% (52.6–56.5)	3195	45.4% (43.5–47.3)	
**Biological characteristics**					
**CD4 cell-count category ^G^**					
<350 cells/µL	641	31.6% (28.8–34.4)	1456	68.4% (65.6–71.2)	<0.0001
350–499 cells/µL	892	52.9% (49.7–55.3)	805	47.1% (43.9–50.3)	
≥500 cells/µL	2717	67.1% (65.0–69.3)	1273	32.9% (30.7–35.0)	
**On ART**					
Yes	3326	84.9% (83.4–86.4)	616	15.1% (13.6–16.6)	<0.0001
No	933	21.9% (20.2–23.6)	2949	78.1% (76.4–79.8)	
**ARV dosage**					
Multiple dose	356	74.0% (68.8–79.1)	125	26.0% (20.9–31.2)	<0.0001
Fixed/single dose	2947	88.7% (87.5–90.0)	387	11.3% (10.0–12.5)	

Missing data for: ^A^, ^B^, ^C^, ^D^ = 14; ^E^ = 10, ^F^ = 127, ^G^ = 40. Missing data excluded from percentage calculation. ZAR = South African Rands (ZAR 15 ~US$1). IQR = Interquartile range. Ever had any STI symptoms = any symptoms of abnormal vaginal discharge, burning, or pain when passing urine, or presence of any genital ulcers/warts. Bold and italics fonts were used to highlight headings and the different characteristics

**Table 2 tropicalmed-07-00232-t002:** Posterior mean and 95% credible interval for the Smooth-term variance components (SVC).

Variables	Posterior Mean	95% Credible Intervals
**Non-linear effect**		
Age	20.69	(18.72, 21.98)
Household size	3618.15	(3305.79, 3872.70)
Number of lifetime HIV tests	14,170.47	(12,778.12, 15,851.12)
**Spatial effect**		
Structured spatial effect	87.97	(70.54, 121.53)
Unstructured spatial effect	27.93	(20.50, 31.57)

**Table 3 tropicalmed-07-00232-t003:** Adjusted odds ratio (AOR) estimates and 95% credible intervals for predictors of unsuppressed HIV viral load among HIV-positive men and women in Vulindlela (rural) and Greater Edendale (peri-urban) areas in the uMgungundlovu District, KwaZulu-Natal Province, South Africa.

Variables	Posterior Mean	Posterior SD	95% Credible Interval
**Year of study** (ref: 2014)			
2015	0.078	0.029	(0.022, 0.135) *
**Gender** (ref: Male)			
Female	0.018	0.011	(0.003, 0.039)
**Educational level** (ref: Completed high school)			
Incomplete high schooling	0.019	0.008	(0.002, 0.035) *
**Marital status** (ref: Never married)			
Ever married	0.012	0.011	(0.009, 0.034)
**Away from home last 12 months** (ref: Yes)			
No	0.043	0.014	(0.016, 0.071) *
**Duration in community** (ref: Always)			
Less than 12 months	0.013	0.022	(0.031, 0.057)
More than 12 months	0.005	0.009	(0.013, 0.023)
**Run out of money last 12 months** (ref: Yes)			
No	0.002	0.012	(0.022, 0.026)
**Had meal cut last 12 months** (ref: No)			
Yes	0.009	0.012	(0.015, 0.034)
**Household monthly Income** (Ref: >R2500)			
≤R2500	0.002	0.010	(0.017, 0.020)
No income	0.015	0.016	(0.015, 0.046)
**Accessing healthcare** (ref: Yes)			
No	0.011	0.009	(0.007, 0.028)
**Alcohol consumption** (ref: No)			
Yes	0.057	0.010	(0.037, 0.077) *
**Knowledge of HIV status** (ref: Positive)			
Negative	0.158	0.017	(0.124, 0.193) *
**Ever tested for HIV** (ref: Yes)			
No	0.050	0.010	(0.031, 0.069) *
**on ART** (ref: Yes)			
No	0.377	0.018	(0.342, 0.412) *
**ARV dosage** (ref: fixed/single)			
Multiple	0.208	0.016	(0.178, 0.239) *
**Ever tested for TB** (ref: No)			
Yes	0.050	0.010	(0.031,0.069) *
**Exposed to TB last 12 months** (ref: No)			
Yes	0.028	0.019	(0.010, 0.066)
**Ever diagnosed with TB** (ref: No)			
Yes	0.008	0.014	(0.020,0.035)
**On TB medication** (ref: No)			
Yes	0.027	0.014	(0.001, 0.055) *
**Perceived risk of contracting HIV** (ref: already infected)			
Likely	0.077	0.015	(0.047, 0.107) *
Not Likely	0.090	0.016	(0.058, 0.122) *
**Forced sex first time** (ref: No)			
Yes	0.006	0.026	(0.046, 0.058)
Don’t remember	0.006	0.023	(0.038, 0.051)
**Number of sex partners last 12 months** (ref: 0–1 partner)			
2 or more partners/No response	0.027	0.011	(0.005, 0.049) *
**Number of lifetime sex partners** (ref: 0–1 partner)			
2 or more partners	0.031	0.019	(0.005, 0.068)
**Current CD4 Cell count** (cells/µL) (ref: < 350 cell/µL)			
350–499 cell/µL	0.161	0.011	(0.138, 0.183) *
≥500 cell/µL	0.270	0.010	(0.251, 0.289) *

Bayesian Geoadditive model. * Implies significant level at 95% CrI. Adjusted for unobserved factors and correlated variables. Bold and italics fonts were used to highlight headings and the different characteristics.

## Data Availability

The datasets generated and analysed during the current study are available from the corresponding author. However, restrictions apply to these data’s availability and are not publicly available due to maintaining participants’ confidentiality and the community involved.

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
