# Peer review of "Spatiotemporal Variation and Predictors of Unsuppressed Viral Load among HIV-Positive Men and Women in Rural and Peri-Urban KwaZulu-Natal, South Africa"

_tropicalmed, 2022, doi:10.3390/tropicalmed7090232_

Round 1

Reviewer 1 Report (Previous Reviewer 2)

The only remark I had about this very interesting article was a remark about the understanding of certain points of it.

So it now appears that this problem has been fixed and therefore this article can be published as is.

Author Response

We would like to thank the reviewer and editor for their time, and effort in reviewing the manuscript which has strengthen our work substantially.

We appreciate the constructive review , corrections , suggestions, and recommendations provided which has added more quality to our manuscript.

We have included a table with a point by point response, location and modified text in response to each of the reviewer's comments.  A track changes version of the revised manuscripts is also attached (changes highlighted in yellow).

Reviewer 2 Report (New Reviewer)

This paper aims to investigate the prevalence, predictors, and high-risk areas of unsuppressed HIV viral load among HIV-positive men and women using the Bayesian geoadditive regression model while accounting for unobserved factors, non-linear effects of selected continuous variables, and spatial autocorrelation using data from a population-based study. Overall, this paper has a nice contribution to the spatial models on modelling HIV. However, some points need to be carved out as follows:

Line 59: “…from AIDS related illness were reported…”

Please write the abbreviation of AIDS because it was first mentioned in the manuscript!

Under Introduction Section, I would suggest providing the research gap.

Line 148: ”… combi PT assay (Roche Diagnostics, Penzberg, Germany…”

Please put a bracket at the end of this sentence!

Line 179: “Geary’s C test was used due to its consistent power…”

Please provide the reference regarding Geary’s C test or provide its formula!

Lines 197-198: “Model comparison was measured using Deviance Inclusion Criteria (DIC),…”

There are many types of DIC. Which DIC did you use? Please provide the reference!

Lines 209-211: “Current age, household size, and the number of lifetime HIV tests were the only non-linear continuous covariates considered in the fitted model, while other continuous covariates were included as linear fixed effects”. Why?

Line 228: “In this case, g denotes a response function and…”

I think there is a” Full stop” before this sentence.

Lines 232 and 244: in equation (2), the Phi symbol denotes the year of study while in equation (4), the Phi symbol denotes the household size. Is that correct?

Lines 246-247: Please provide more explanation regarding equation (4)!

Lines 281-282: Please revise the sentence!

Line 461, Table 2: You provide DIC saturated, but there is no explanation regarding DIC saturated. What is DIC saturated?

Lines 466-469 and Table 3: “The implication of this is the lower the SVC, the more accurate the model. The structured spatial effect shows SVC of 87.97 which was higher compared to that of the unstructured spatial effect of 27.93,…”

However, in Table 3, the SVC of Age is 20.69. How did you explain this result? Please provide more explanation regarding the “Non-linear effect” as well!

Lines 630-635: Why you did not mention “Perceived risk of contracting HIV” as a significant predictor?

Author Response

We would like to thank the reviewer and editor for their time, and effort in reviewing the manuscript which has strengthen our work substantially.

We appreciate the constructive review, corrections, suggestions, and recommendations provide which has added more quality to our manuscript.

We have included a table with a point-by-point response, location , and modified text in response to each of the reviewers comments. A track changes version of the revised manuscripts is also attached ( changes made are highlighted in yellow

Reviewer 3 Report (New Reviewer)

Title "Spatial Variation and Predictors of Unsuppressed Viral Load Among HIV Positive Men and Women in Rural and Peri-urban KwaZulu-Natal, South Africa"

The authors investigated the prevalence, predictors, and high-risk areas of unsuppressed HIV viral load among HIV positive men and women, using Bayesian Geoadditive modelling.

This is an interesting topic analyzed with a relevant approach. The research objectives have been achieved. The title is related to spatial variation, so the data should have been analyzed according to the type of enumeration area (peri-urban vs. rural) to seek factors’ variation taking according to enumeration areas type. But this last suggestion doesn’t affect the current research quality.

The following suggestions should be addressed to improve the paper quality.

Hasn't this study already been published? https://www.intechopen.com/online-first/82367

WRITTEN BY: Adenike O. Soogun, Ayesha B.M. Kharsany, Temesgen Zewotir and Delia North;  Submitted: May 2nd, 2022 Reviewed: May 25th, 2022 Published: June 24th, 2022

Many statistical tests are used.  But their contribution to the achievement objectives is not always explained in the methodology; Some have already been published by the authors and do not necessarily need to be reused here. For example, "Multiple Corresponding 172 Analysis (MCA) and Random Forest Analysis (RFA)". Line 172 - 173.

The choice of spatial autocorrelation tests should be well explained.

Line 126 (Figure 1): Use a visible outline for the boundary between Vulindlela and Greater Edendale on the map and put the name labels on the map.

Lines 139 to 140 : The 15 were excluded from the 7839. So rephrase the sentence to be clear.

Line 173 : Findings in a previous analysis of this study database or "in the literature"?

Lines 178 to 180: It’s not clear. Did you first do a Moran global autocorrelation test to see if the variable distribution is random, concentrated, or homogeneous before doing Geary’s C tests?

Lines 215 to 216: Were descriptive and bivariate statistical analyses carried out with SAS software and the modeling with R software? If so, please indicate it here. It seems that the Figure 4 map was made with R software. If so, precise it in the methodology.

Lines 266 to 267: The three statistical models comparison is not relevant, given that the Unstructured Model is the preferred model, section 3.6 and Table 2 could be moved into supplementary materials.

Lines 287 to 288: To be moved to the methodology.

Line 432: When talking about Hotspot, Local Indicators of Spatial Association (LISA) should be interested (Local Indicators of Spatial Association—LISA;  https://doi.org/10.1111/j.1538-4632.1995.tb00338.x) .  Or the local, Geary autocorrelation ("A Local Indicator of Multivariate Spatial Association: Extending Geary's c. of Luc Anselin. DOI:10.13140/RG.2.2.18101.58084)

Line 438: The maps (Figure 2 and Figure 4)  layout  needs to be reviewed. Are these study areas with prevalence <400 copies/ml ? Use a legible line outline to materialize the boundary between Vulindlela and Greater Edendale on the map and put these two names (Vulindlela and Greater Edendale ) labels on the map.

Lines 444  to 445: This should appear in the methodology, point "2.5. Statistical analysis".

Lines 453 to 463: To move in Supplementary Materials

Line 591: Figure 4. It seems that this map was made with R software. If so to be specified in the methodology. Review the layout of the map.

Lines 733 to 734 : In the text, the enumeration areas have not been classified into 3 classes.

Supplementary material: Table 1 S1. Place of residence Peri-urban instead of Urban

Supplementary material: Table 2 S2. Enumeration area Peri-urban instead of Urban

Author Response

We would like to thank the reviewer and editor for their time, and effort in reviewing the manuscript, which has strengthened our  work substantially.

We appreciate the constructive review, corrections, suggestions, and recommendations provided which has added more quality to our manuscript.

We have included a table with a point-by-point response, location and modified text in response to each of the reviewers comments. A track changes version of the revised manuscripts is also attached with highlighted changes in yellow.

Reviewer 4 Report (New Reviewer)

General comments:

This paper proposes an interesting discussion on predictors for unsuppressed viral load, while accounting for spatial variation, in South Africa, which has the world’s largest HIV epidemic. While the Geoadditive models presented in this paper are not novel in the literature, there is some merit of the proposed application to the HIV data.

Minor issues:

The mathematical notation used in the statistical methods section should be clear and more consistent.

-      Line 227: please explain what  stands for.

-      Line 228: the use of ,  and  is not consistent. There are other instances in the text where they have different meanings.

-          Line 229: missing the word “continuous”? “…comprise spatial, continuous and categorical … “

-          Eq (5): please, specify which spatial structured factors (random effects) were introduced to the model?

-      Eq (2) to (6): use of xijm and  is not introduced. Subscripts m and k need to be explained.

-          Eq (7): consider adding a comma to separate the notation for the mean and the variance

-          Line 295: what is the data and what are the parameters? Note that the posterior is the probability of the parameters given the data. Is x a parameter?

-          Which prior distributions were assumed for the hyper-parameters?

Author Response

We would like to thank the reviewer and editor for their time, and effort in reviewing the manuscripts, which has strengthen  our work substantially.

We appreciate the construction review, corrections, suggestions and recommendations provided which has added more quality to our manuscript.

We have included a table with a point-by-point response, location and modified test in response to each of the reviewer's comments. A track changes version of the revised manuscript is also attached with changes highlighted in yellow

Thank you 

This manuscript is a resubmission of an earlier submission. The following is a list of the peer review reports and author responses from that submission.

Round 1

Reviewer 1 Report

in the introduction, the authors do not explicitly express what is to be gained when advanced spatial methods are used to model unsupressed HIV viral load. They also do not make it clear the gains when spatial effects is incorporated in the model. 

I find the topic very important; however, the description of the methods lack clarity. There are so many statistical analysis conducted but it is not clear what their value is. Typically, the authors at some point is using viral load as the response variable and at some point also uses the binary outcomes of the test results. I find it troubling for potential readers. 

I suggest the authors completely rewrite the methods section focusing on the important statistical methods useful to address their research objectives 

Author Response

Kindly find point by point response to Reviewer comments

Reviewer 2 Report

First of all, I have to say that I found this study very interesting, as it deals with a burning global health issue on a continent where the problem is really huge. Therefore, the thought of the respected authors was aimed at people who have great difficulty accessing health care. This fact alone deserves everyone's attention.

From the point of view of methodology as well, and in terms of spatial statistics, I think that the authors have worked with a very good and innovative approach, and their effort should be an example, especially for diseases with a multifactorial character.

They used a lot of bibliographic references and structured their text with scientific completeness.

So I would have no objection to suggesting the immediate publication of the article. However, I think that the authors should revisit the wording of the text a bit, as there are several grammatical and syntactic errors, which in some cases make it difficult to understand.

Best regards

Author Response

Kindly find attached point by point response  to reviewer's comments and report 

Round 2

Reviewer 1 Report

Although the topic is relevant, I find the content of the manuscript haphazardly structured. There is no significant improvement in relation to the previous version. I rather see an important topic, with many statistical methods applied but unrelated to the objectives. In the nutshell, the manuscript lacks clarity. The author should address the following specific contents if the want to get their paper published in the future 

Lines 180-182: On which variable was the Geary's C test applied? Recall that your response variable is binary. Does this mean you aggregate the response over the EAs? If so, then kindly make it clear.

Lines 185-187. This sentence is very confusing

Lines 192-194: The outcome of interest is still not clear in the manuscript. Are the authors referring to the binary classifications; i.e. y=suppressed when viral load <400 and y = unsuppressed otherwise.

lines 196: How were the nonlinear effects of continuous covariates examined?

Lines 212-213: This statement lacks clarity. There are 5 variables in the brackets. Which of them represents the response variable?

line 222: I don’t know what the authors mean by non-linear spatial effect

Lines 242-243: There is a lack of consistency. In equation 1, the variable ς is used to represent age; but in equation 4, the same variable is used to represent structured spatial random effects.

Line 255: The author should give a reason why the final discussion is based on the third model.

Lines 259-260: It is not clear why the author used Gaussian random field to study the spatial structure of the variable via semi-variogram (that variable is still even unclear) but later chose to use Gaussian Markov random field as prior (CAR prior) for the spatial effects. Also, the spatial entities on which the spatial prior is based is not clear in the model formulation. Specifically, in line 213, the authors use i to represent districts but in the material and methods section, the authors use enumeration areas (EAs) as their spatial entities.

lines 368-370: The authors state that the Gaussian semi-variogram model is the most appropriate structure for their data. But the parameters or graph of this model is not shown. How can a reader deduce the structure without the parameters? Besides, the relevance of the semi-variogram to their further analysis, for example, the spatial modeling, is missing. The authors have then turned to the Gaussian Markov prior instead of the Gaussian random field prior based on their semi-variogram.